# The Efficacy of Transcranial Direct Current Stimulation in Enhancing Surgical Skill Acquisition: A Preliminary Meta-Analysis of Randomized Controlled Trials

**DOI:** 10.3390/brainsci11060707

**Published:** 2021-05-27

**Authors:** Chao-Ming Hung, Bing-Yan Zeng, Bing-Syuan Zeng, Cheuk-Kwan Sun, Yu-Shian Cheng, Kuan-Pin Su, Yi-Cheng Wu, Tien-Yu Chen, Pao-Yen Lin, Chih-Sung Liang, Chih-Wei Hsu, Che-Sheng Chu, Yen-Wen Chen, Ming-Kung Wu, Ping-Tao Tseng

**Affiliations:** 1Division of General Surgery, Department of Surgery, E-Da Cancer Hospital, Kaohsiung 82445, Taiwan; ed100647@edah.org.tw; 2School of Medicine, College of Medicine, I-Shou University, Kaohsiung 82445, Taiwan; 3Department of Internal Medicine, E-Da Hospital, Kaohsiung 82445, Taiwan; holdinggreat@yahoo.com.tw (B.-Y.Z.); b95401072@ntu.edu.tw (B.-S.Z.); 4Department of Emergency Medicine, E-Da Hospital, Kaohsiung 82445, Taiwan; lawrence.c.k.sun@gmail.com; 5School of Medicine for International Students, College of Medicine, I-Shou University, Kaohsiung 82445, Taiwan; 6Department of Psychiatry, Tsyr-Huey Mental Hospital, Kaohsiung Jen-Ai’s Home, Kaohsiung 831, Taiwan; n043283@gmail.com; 7Mind-Body Interface Laboratory (MBI-Lab), China Medical University Hospital, Taichung 40447, Taiwan; cobolsu@gmail.com; 8College of Medicine, China Medical University, Taichung 40447, Taiwan; 9An-Nan Hospital, China Medical University, Tainan 709, Taiwan; 10Department of Sports Medicine, Landseed International Hospital, Taoyuan 32449, Taiwan; Keanu.firefox@gmail.com; 11Department of Psychiatry, Tri-Service General Hospital, Taipei 11490, Taiwan; verducciwol@gmail.com; 12School of Medicine, National Defense Medical Center, Taipei 11490, Taiwan; lcsyfw@gmail.com; 13Institute of Brain Science, National Yang Ming Chiao Tung University, Taipei 112, Taiwan; 14Department of Psychiatry, Kaohsiung Chang Gung Memorial Hospital and Chang Gung University College of Medicine, Kaohsiung 83301, Taiwan; paoyenlin@gmail.com (P.-Y.L.); harwicacademia@gmail.com (C.-W.H.); 15Institute for Translational Research in Biomedical Sciences, Kaohsiung Chang Gung Memorial Hospital and Chang Gung University College of Medicine, Kaohsiung 83301, Taiwan; 16Department of Psychiatry, Beitou Branch, Tri-Service General Hospital, School of Medicine, National Defense Medical Center, Taipei 11490, Taiwan; 17Graduate Institute of Medical Sciences, National Defense Medical Center, Taipei 11490, Taiwan; 18Department of Psychiatry, Kaohsiung Veterans General Hospital, Kaohsiung 813, Taiwan; youngtzuchi@hotmail.com; 19Center for Geriatric and Gerontology, Kaohsiung Veterans General Hospital, Kaohsiung 813, Taiwan; 20Non-invasive Neuromodulation Consortium for Mental Disorders, Society of Psychophysiology, Taipei 11490, Taiwan; kevinachen0527@gmail.com; 21Prospect Clinic for Otorhinolaryngology & Neurology, Kaohsiung 811, Taiwan; 22Institute of Biomedical Sciences, National Sun Yat-sen University, Kaohsiung 80424, Taiwan; 23Department of Psychology, College of Medical and Health Science, Asia University, Taichung 40704, Taiwan

**Keywords:** surgical skill, resident training, meta-analysis, tDCS, neuromodulation

## Abstract

The application of transcranial direct current stimulation (tDCS) to targeted cortices has been found to improve in skill acquisition; however, these beneficial effects remained unclear in fine and complicated skill. The aim of the current meta-analysis was to investigate the association between tDCS application and the efficacy of surgical performance during surgical skill training. We included randomized controlled trials (RCTs) investigating the efficacy of tDCS in enhancing surgical skill acquisition. This meta-analysis was conducted under a random-effect model. Six RCTs with 198 participants were included. The main result revealed that tDCS was associated with significantly better improvement in surgical performance than the sham control (Hedges’ *g* = 0.659, 95% confidence intervals (95%CIs) = 0.383 to 0.935, *p* < 0.001). The subgroups of tDCS over the bilateral prefrontal cortex (Hedges’ *g* = 0.900, 95%CIs = 0.419 to 1.382, *p* < 0.001) and the primary motor cortex (Hedges’ *g* = 0.599, 95%CIs = 0.245 to 0.953, *p* = 0.001) were both associated with significantly better improvements in surgical performance. The tDCS application was not associated with significant differences in error scores or rates of local discomfort compared with a sham control. This meta-analysis supported the rationale for the tDCS application in surgical training programs to improve surgical skill acquisition.

## 1. Introduction

At least 321 million surgical procedures were performed to treat human diseases in 2010 [1]. Advanced surgical skill acquisition requires high-intensity training programs and high workloads [2]. However, such extreme high-intensity workloads could result in an increased risk of error, which can contribute to serious morbidity and mortality in patients [3,4]. Therefore, the adjustment of the workload of surgical residents (i.e., duty-hour restrictions) has become a dilemma, reducing the workload but also limiting opportunities for trainees to gain proficiency [5]. A previous meta-analysis demonstrated no benefit in duty-hour restrictions in the improvement of safety or reductions in morbidity/mortality of patients receiving surgical procedures; thus, the accommodation of resident training needs has been recommended [6].

Several new techniques have been applied to improve surgical skill training programs, such as simulation-based task training [7]. Simulation-based task training had the advantage of being low risk and effective in helping trainees attain surgical skills [8]. However, there are still several disadvantages in acquiring complex surgical motor skills through a simple simulation-based task training program, including rapid skill decay, and the program is time-consuming and only modestly effective [9,10,11]. Therefore, it is important to identify an alternative to enhance the efficacy of simulation-based task training.

Transcranial direct-current stimulation (tDCS) may improve both the executive function and motor-learning function in healthy volunteer through brain function modulation [12], which two functions were the key components in surgical skill acquisition/training. The potential beneficial effect on executive function might be derived from its enhancing effect on long-term potentiation and brain-derived neurotrophic factor expression in the stimulated brain regions [13]. The motor function improving effect might be contributed by increasing or decreasing neuronal activity based on stimulation polarity and its current intensity [14,15,16,17]. For example, the previous articles had summarized the evidence of tDCS efficacy in motor skill training (either in fine movement or reaction time) in stroke patients and healthy subjects [18,19]. To be specific, Karok and the colleagues had demonstrated different electrode montage-dependent improvement in three experimental tasks (i.e., Purdue Pegboard test, visuomotor grip force tracking task, and visuomotor wrist rotation speed control task) by tDCS application in healthy young adults [20]. Previous randomized controlled trials (RCTs) have found significant beneficial effects in surgical performance during surgical training [10,21]. However, these beneficial effects were not consistent among the different surgical training programs [9]. In addition, differences in the targeted cortex result in different results (i.e., targeted at the primary motor cortex vs. at the supplementary motor area) [22]. Furthermore, although many meta-analyses have demonstrated the benefit of tDCS in the improvement of gross motor activities [23,24], no previous meta-analysis has specifically investigated the benefit of tDCS in surgical skill acquisition, which was recognized to involve complex visuospatial memory, executive function, attention, and fine movement [10].

In order to provide a point of view of potentially beneficial effect by tDCS application in surgery skill acquisition, the aim of the current preliminary meta-analysis was to investigate the association between the application of tDCS and the efficacy of surgical performance during surgical skill training. Furthermore, in addition to the overall result of tDCS application, we made a subgroup analysis based on the different stimulation cortical sites in order to determine the potentially different effects by the different stimulation sites. Finally, in order to provide as many information as possible, we also intended to include all different non-invasive brain stimulation method in our literature search strategy.

## 2. Materials and Methods

The detailed information had been descripted in Appendix A. In brief, the current meta-analysis followed the latest PRISMA 2020 guidelines (Appendix A) [25] and AMSTAR2 (assessing the methodological quality of systematic reviews) guidelines [26]. The current study complies with the Institutional Review Board of the Tri-Service General Hospital (TSGHIRB: B-109-29). This study had been registered in INPLASY202140099 (https://inplasy.com/inplasy-2021-4-0099/, accessed on 19 April 2021); DOI number: 10.37766/inplasy2021.4.0099. Electronic searches with the keyword of (deep transcranial magnetic stimulation OR dTMS OR repetitive transcranial magnetic stimulation OR rTMS OR TMS OR non-invasive brain stimulation OR theta burst stimulation OR transcranial direct current stimulation OR TBS OR tDCS OR vagus nerve stimulation OR vagal nerve stimulation OR tVNS OR nVNS OR VNS OR static magnetic field stimulation) AND (skill OR professionalism OR skill acquisition) AND (surgery OR surgical OR surgeon) AND (random OR randomized OR randomised) in the PubMed, Embase, ClinicalKey, Cochrane CENTRAL, ProQuest, ScienceDirect, Web of Science, and ClinicalTrials.gov platforms through 19 April 2021 had been conducted (the detailed search strategy is provided in Appendix A). Furthermore, to expand the pool of potential studies, we consulted the reference lists of review articles and performed further manual searches [27,28,29]. No language restriction had been applied.

The PICO (population, intervention, comparison, outcome) setting of the current meta-analysis included: (1) P: participants receiving surgical skill training; (2) I: transcranial direct-current stimulation (tDCS); (3) C: sham stimulation or active control; and (4) O: the change in surgical performance. Only RCTs, either sham-control or active control, investigating the difference in changes of surgical performance after tDCS or sham stimulation had been included. The methodological quality of recruited studies was evaluated with the Cochrane risk-of-bias tool [30].

The primary outcome was the change in surgical performance associated with tDCS or sham stimulation. Because the definition of surgical performance varied widely among the individual RCTs, we did not specifically set a limitation to the specific definition of surgical performance. Rather, we chose the primary outcome of surgical performance applied in each RCT to represent the primary findings of each RCT. Secondary outcomes were the changes in error scores. The error scores were defined as improper transfers in laparoscopic training or resection of healthy brain in neurosurgical training. Safety profile was defined as the rate of local discomfort (i.e., itching, tingling pain, or erythematous).

Based on the presumed heterogeneous selection population among all the recruited studies, the current meta-analysis was conducted with random-effects meta-analysis models. The meta-analysis procedure was performed with Comprehensive Meta-Analysis software, version 3 (Biostat, Englewood, NJ, USA). We chose Hedges’ *g* and 95% confidence intervals (95%CIs) as the main effect sizes (ESs) of the primary and secondary outcomes. We chose odds ratios (ORs) and 95%CIs as the ESs of the safety profiles. To detect potential heterogeneity, we used the *Q* statistic and corresponding *p* values [31]. At the same time, to evaluate the potential publication bias, we visually inspected funnel plots [32] or performed Egger’s regression tests [33]. We used the Duval and Tweedie trim-and-fill test when there was evidence of publication bias [34]. A sensitivity test was performed using the one study removal method, in which one study was excluded from the analyses at a time to observe whether significant or insignificant results of the meta-analyses were biased by outliers [35]. We used the meta-regression procedure and subgroup meta-analyses to discover the potential sources of heterogeneity and confounding factors. In addition, we performed a subgroup meta-analysis according to the cortex at which the tDCS was targeted. In addition, we examined the differences in the ESs of individual subgroups with interaction tests [36]. Furthermore, to confirm the reliability of the result of the current meta-analysis, we made further subgroup analysis according to subgroup of meeting abstracts or subgroup of formally published articles.

## 3. Results

### 3.1. Study Selection

The protocol for study selection in the current meta-analysis is depicted in Figure 1. In brief, twelve full-text articles were eligible. Among them, six were excluded because they did not fulfill the inclusion criteria (Appendix A). Therefore, six articles were retained for the current meta-analysis (Table 1) [9,10,21,22,37,38].

Across the 6 eligible RCTs, a total of 198 participants with a mean age of 23.7 years (range 21.6–25.7) and a female proportion of 62.6% (range 46.7–72.7) were included. Among these six eligible RCTs, three investigated the efficacy of anodal tDCS over dominant primary motor cortex/cathodal tDCS over contralateral hemisphere surrounding F3/F4 for 20 min [9,10,21], one investigated the efficacy of cathodal tDCS over right primary motor cortex (M1) stimulation and anode over left primary motor cortex (M1) [22], one investigated the efficacy of tDCS without detailed information about the specific polarity over prefrontal cortex stimulation [37], one investigated anodal tDCS over left dorsolateral prefrontal cortex (DLPFC) (F3) and cathodal tDCS over right DLPFC (F4) [38], and one investigated the efficacy of tDCS over supplementary motor area stimulation (the anode over Cz and the cathode over Fpz) [22]. All the tDCS stimulation had been applied during their initial training session. The current intensity and stimulation duration in each session ranged between 1–2 mA and 15–20 min. There were not any RCTs investigating the efficacy of other non-invasive brain stimulation on surgical skill training program. All the included RCTs were double-blind design and applied a simulation training program as their tool for surgical skill training. The target surgical skill to acquire/train across the included RCTs included knot-tying or suturing task, fundamentals of laparoscopic surgery, peg transfer tasks, and ultrasonic aspiration of virtual tumors (Table 1). The investigated surgical skill performance was rated by pattern-cutting scores, percent of changes in the amount of tumor resected, knot tensile strength, and overall performance score. Because most included RCTs provided results for “immediately after completion of training” but not results for “at follow-up after completion of training”, we chose to extract the data for “immediately after completion of training”.

### 3.2. Methodological Quality of Included Studies

We found that 73.8% (31/42 items), 16.7% (7/42 items), and 9.5% (4/42 items) of the included studies had low, unclear, and high risks of bias, respectively. The vague reporting of concealment contributed to the major source of “unclear risk of bias” (Appendix A).

### 3.3. Primary Outcome: Changes in Surgical Performance

Among six eligible articles that provided datasets on changes in surgical performance [9,10,21,22,37,38], the main result of the meta-analysis revealed that tDCS was associated with significantly better improvement in surgical performance than the sham control (*k* = 6, Hedges’ *g* = 0.659, 95%CIs = 0.383 to 0.935, *p* < 0.001) (Figure 2A), without significant heterogeneity (*Q* value = 4.808, df = 5, *p* = 0.440; *I*^2^ < 0.001%) but with significant publication bias via inspection of the funnel plot (Appendix A). The adjusted ES after the trim-and-fill test remained significant (Hedges’ *g* = 0.607, 95%CIs = 0.326 to 0.887, *p* < 0.001).

#### 3.3.1. Sensitivity Test

The sensitivity test via one study removal examination revealed that the main results of the meta-analysis did not change after removing any one of the included datasets.

#### 3.3.2. Meta-Regression

The meta-regression procedure showed that none of the investigated clinical variables was significantly associated with the ES, including mean age (*p* = 0.662), female proportion (*p* = 0.376), or intensity of electrical current applied in tDCS (*p* = 0.718) (Appendix A).

#### 3.3.3. Subgroup Meta-Analysis: Different Targeted Cortices

We performed a further subgroup meta-analysis according to the different cortices targeted by tDCS. The bilateral prefrontal cortex (*k* = 2, Hedges’ *g* = 0.900, 95%CIs = 0.419 to 1.382, *p* < 0.001) and primary motor cortex (*k* = 4, Hedges’ *g* = 0.599, 95%CIs = 0.245 to 0.953, *p* = 0.001) subgroups showed similar results, indicating that tDCS was associated with significantly better improvement in surgical performance than the sham control. However, there was no significant difference in changes in surgical performance with tDCS over the supplementary motor area than with the sham control (*k* = 1, Hedges’ *g* = 0.176, 95%CIs = −0.433 to 0.785, *p* = 0.571) (Figure 2B). There was no significant difference between the ESs of these three subgroups according to the interaction test (*p* = 0.187).

#### 3.3.4. Subgroup Meta-Analysis: Meeting Abstracts or Formally Published Articles

We performed a further subgroup meta-analysis according to the publication types. The subgroup of meeting abstract (*k* = 1, Hedges’ *g* = 0.982, 95%CIs = 0.242 to 1.721, *p* = 0.009) and subgroup of formally published articles (*k* = 5, Hedges’ *g* = 0.607, 95%CIs = 0.310 to 0.904, *p* < 0.001) both showed similar results, indicating that tDCS was associated with significantly better improvement in surgical performance than the sham control (Appendix A).

### 3.4. Secondary Outcome: Changes in Error Score

Three eligible articles provided 3 datasets of changes in error score [10,22,38]. In brief, the main result of the meta-analysis revealed that tDCS was not associated with significantly different changes in error score compared with the sham control (*k* = 3, Hedges’ *g* = 0.460, 95%CIs = −0.183 to 1.102, *p* = 0.161) (Figure 2C), without significant heterogeneity (*Q* value = 5.756, df = 2, *p* = 0.056; *I*^2^ = 65.252%).

#### Subgroup Meta-Analysis: Different Targeted Cortices

We performed a further subgroup meta-analysis according to the different cortices targeted by tDCS. The subgroup of primary motor cortex (*k* = 2, Hedges’ *g* = 0.083, 95%CIs = −0.403 to 0.568, *p* = 0.739) showed similar results, i.e., that tDCS was not associated with significantly different changes in error scores compared with the sham control. However, there was a significantly better improvement in error score with tDCS over the supplementary motor area (*k* = 1, Hedges’ *g* = 0.758, 95%CIs = 0.128 to 1.388, *p* = 0.018) or over bilateral prefrontal cortex (*k* = 1, Hedges’ *g* = 1.105, 95%CIs = 0.451 to 1.759, *p* = 0.001) than with the sham control (Figure 2D). There was a significant difference between the ESs of these two subgroups according to the interaction test (*p* = 0.034).

### 3.5. Safety Profile: Rates of Local Discomfort (i.e., Itching, Tingling Pain, or Erythematous)

There were 3 eligible articles providing 3 datasets of safety profiles [9,21,22]. In brief, the main results of the meta-analysis revealed that tDCS was not associated with significantly different safety profiles regarding aspects of local discomfort compared with the sham control (*k* = 3, OR = 2.108, 95%CIs = 0.796 to 5.587, *p* = 0.134) (Figure 2E), without significant heterogeneity (*Q* value = 0.971, df = 2, *p* = 0.615; *I*^2^ < 0.001%). To be specific, the most frequently reported adverse event included tingling, burning sensation, itching, and pain (Table 1).

## 4. Discussion

To our knowledge, this is the first meta-analysis addressing the efficacy of tDCS in improving surgical performance during surgical training. The main result of the current preliminary meta-analysis revealed that tDCS was associated with significantly better improvement in surgical performance. In addition, the significant beneficial effect on surgical performance by tDCS mainly focused on the active tDCS targeted at the bilateral prefrontal cortex or the primary motor cortex but not the supplementary motor area. Finally, in terms of safety, the application of tDCS was not associated with significantly different error scores or rates of local discomfort compared with the sham control.

The main finding of the current study was that tDCS was associated with significantly better improvement in surgical performance but not error scores than the sham control. The significantly better improvement in the tDCS groups than those in the sham controls would suggest that the application of tDCS would be associated with not only simply placebo effect but also potential beneficial effect in the surgical skill acquisition. The investigated tDCS in the current study included anode tDCS 1 mA at the dominant primary motor cortex + cathode at the contralateral supraorbital area [9,10], anode tDCS 1 mA at the dominant primary motor cortex (C3/C4) + cathode at contralateral F3 or F4 [21], cathodal tDCS 2 mA at the right primary motor cortex (C4) + anode at the left primary motor cortex (C3) [22], cathodal 2 mA above the bridge of the nose (Fpz) and anode at Cz (targeted for supplementary motor area) [22], tDCS 2 mA targeted at the bilateral prefrontal cortex without detailed information about the specific polarity of the tDCS [37], and anodal tDCS over left dorsolateral prefrontal cortex (DLPFC) (F3) plus cathodal tDCS over right DLPFC (F4) [38]. tDCS has been found to have an enhancing effect on the targeted cortex via anode current [14,15]; however, the suppressing or enhancing effect due to the cathodal tDCS is still debated [16,17]. To be specific, although there was inconclusive evidence the low-current (i.e., 1 mA) cathodal tDCS stimulation might contribute to the suppressing effect [39], the high-current (i.e., 2 mA) cathodal tDCS stimulation might be associated with excitatory changes [40]. Therefore, all the investigated tDCS stimulations included in the current study might all have enhancing effects on the targeted cortex (i.e., anodal tDCS 1 mA or cathodal tDCS 2 mA).

The enhancing effect of tDCS on the targeted cortex has been found to improve participants’ perception, cognitive functioning, and motor function [14]. Stimulation site over the prefrontal cortex would result in improvement in both the executive function and motor-learning function in healthy volunteer through brain function modulation [12]. Similarly, the application of tDCS enhancement over participants’ primary motor cortex was found to be associated with significant improvements in balance control in stroke patients and healthy subjects [18,23]. In the study by Karok, the tDCS stimulation over M1, either right M1 or bilateral M1, significantly improved three fine movement-related task, including Purdue Pegboard test, visuomotor grip force tracking task, and visuomotor wrist rotation speed control task, compared to sham control [20]. All these key components (i.e., executive function, motor-learning function, fine movement, and reaction time) were the major part of fine movement skill acquisition; therefore, the improvement of these key components would support the rationale of the application of enhancing tDCS to improve motor skill acquisition (i.e., surgical skill acquisition) [10,41]. Alterations in excitability by the weak electric field in neurons [39] might be one of the reasons explaining its effect on skill acquisition. After the application of enhancing tDCS and the consequent electric field over the primary motor cortex, tDCS provides short-term modulatory effects on electroencephalography (EEG) composition, especially on delta- and alpha-frequency waves, in the participants’ sensorimotor regions, which were associated with the surgical training and the performance of unimanual and bimanual skills [21]. In the RCT by Ciechanski, P. (2019) [21], the enhancing tDCS would significantly improve the surgical skill performance during follow-up period, which might be associated with the consolidation of a continuous skill task and might help to prolong the efficacy of skill acquisition enhancement [41]. In the previous rat study, the acquisition of a specific skill after 1 day of training should rely on plasticity-related protein synthesis [42]. Therefore, the beneficial effect of enhancing tDCS on skill acquisition might result from direct enhancement of protein synthesis during training or an indirect modulatory effect on the downstream learning-related protein synthesis [41]. However, the current study could not provide further evidence to determine which mechanism tDCS relied to induce the consolidation. Therefore, future studies specifically targeted at the effects of tDCS on memory consolidation are warranted.

Another concern about the application of tDCS-assisted surgical skill training is the safety and potential risk of an excitatory/inhibitory effect over the unwanted cortex. In most RCTs, the anode/cathode electrodes were large enough to result in a nonfocal electric field, which was likely to widely spread to the nearby cortical or unwanted subcortical regions [9], which might cause the effect of tDCS to be unpredictable. In addition, the safety profile and tolerability were among the major ethical concerns when applying an intervention to healthy subjects. In the main result of the current study, the application of tDCS in the healthy participants was not associated with significantly different error scores (*p* = 0.161) or rates of local discomfort (*p* = 0.134) compared with the sham control. Although we did not perform an analysis of acceptability (i.e., drop-out rate), because only one RCT provided such information, no significantly different drop-out rates were noted between the tDCS and sham groups in that RCT [22]. Furthermore, there were no serious adverse events (i.e., seizure or mortality) noted in the included RCTs. The most frequently reported adverse event were relatively mild and included tingling, burning sensation, itching, and pain (Table 1). Therefore, the current meta-analysis supports the safety and tolerability of the application of tDCS in surgical skill training.

Another issue regarding the efficacy of tDCS application in skill training program was the “actual effect” of the applied tDCS on brain excitability. For example, the applied current intensity of tDCS would be one major argument. To be specific, the argument of the enhancing/suppressing effect to the target cortex by different current intensity (i.e., low-current versus high-current) had become one inconclusive issue [39,40]. Further, in the previous review article by Esmaeilpour and the colleague, the authors analyzed the dose-response association of tDCS intensity and found a complex state-dependent non-monotonic dose response according to human neurophysiology studies [43]. This finding would further complicate the studies of tDCS in human brain function. In addition, the different targeted cortices would yield different results (i.e., targeted to the primary motor cortex vs. to the supplementary motor area). There had been argument about the different cognition effect of tDCS on differently targeted cortex [44]. However, in the previous review of computational neuroscience modeling studies on stochastic resonance, the “neural noise” produced by tDCS, either in forms of depolarization or hyperpolarization in different cortex, can indeed improve cognitive performance [44,45]. To be specific, according to the neural noise hypothesis, the after-effect of tDCS might depend on the overall glutamatergic, GABAergic, dopaminergic, and serotoninergic synaptic activity. Therefore, the improvement of cognition should not be simply attributed to “better activity, better function” [44]. Therefore, the potentially beneficial effect of tDCS on cognitive performance might not be simply explained by the polarity or targeted cortex of tDCS. Future RCTs investigating different efficacy on surgical skill acquisition by tDCS over different targeted cortex should be warranted.

Finally, in the previous reports, the application of enhancing tDCS over a unilateral motor cortex was beneficial in unimanual tasks [9,10,21], which might not completely address the real-world surgery need. In a previous randomized controlled trial of healthy participants, bilateral tDCS was found to improve bimanual coordination in simple motor training [46]. In complex motor training, such as training involving surgical skills, the application of enhancing tDCS over bilateral motor cortex has been found to have a benefit on bimanual surgical skill learning and shortening the necessary time to achieve a satisfactory skill level [22]. However, because there was only one RCT investigating the efficacy of tDCS over the bilateral motor cortex on the unimanual/bimanual function [22], we could not perform further subgroup analysis focusing on the changes in unimanual/bimanual function by bilateral vs. unilateral stimulation. The number of the RCT investigating the efficacy of tDCS over the bilateral motor cortex or supplementary motor area on the unimanual/bimanual function was relatively smaller than the other subgroups. Therefore, clinicians should pay special attention when applied our result in their surgical training program. Future RCTs focusing on the efficacy of tDCS over the bilateral motor cortex should explore the potential benefits for unimanual/bimanual function.

There were some limitations in the current study. First, there were only small numbers of RCTs and participants (6 RCTs and a total of 198 participants) included in the current study. Some subgroup analyses only consisted of one to two RCTs. However, the difficulty to recruit large numbers of surgeons to join in such RCT would limit the overall numbers of such kinds of RCTs. Therefore, it would be difficult to recruit large numbers of RCTs in the meta-analysis investigating interventions to enhance surgical skill training program. Second, a wide variety of surgical simulation and variety of evaluation scales for surgical performance measurement, such as pattern-cutting scores, percent of changes in the amount of tumor resected, knot tensile strength, and overall performance score, was applied among the included RCTs. The choice of the surgical performance measurement in each RCT varied along with the surgery type (i.e., neurosurgery or laparoscopic surgery) and the method of surgery simulation. Although we could not perform specific subgroup analyses based on specific surgical performance measurements, there had been significant improvement in pattern-cutting scores, percentage of changes in the amount of tumor resected, knot tensile strength, and overall performance score according to the result of their original RCTs. Although there was no statistically significant heterogeneity noted in the current study, the clinicians should pay special attention when applying the results of the current study in their surgical training program. Third, some RCTs applied subgroups of “low skill” and “high skill” surgeons to distinguish the potentially different efficacies of tDCS on surgical performance [10,21]. However, we did not perform such a subgroup analysis in the current study because there was no consensus for defining “low skill” or “high skill” surgeons. Fourth, we extracted the results of “immediately after completion of training” but not the results of “follow-up after completion of training” because only a few RCTs provided such information. Therefore, we could not provide further information about the effects of tDCS in the long-term follow up. Fifth, although we tried to include the other non-invasive brain stimulation, such as repetitive transcranial magnetic stimulator or theta burst stimulation, in the current study by adding such keywords in our search strategy, we did not retrieve sufficient RCTs because no publications were available. Finally, the study design and tDCS protocols applied in the included RCTs were widely heterogeneous (i.e., number of tDCS sessions, duration of each session, electrodes shape/size, type of medium used (for example, normal saline), stimulation intensity, method used to localize the cortical areas (for example 10–20 EEG system or Neuronavigation), and type of sham).

## 5. Conclusions

This current preliminary meta-analysis supported the rationale of the application of enhancing tDCS in a surgical training program to improve surgical skill performance, including pattern-cutting scores, percent of changes in the amount of tumor resected, knot tensile strength, and overall performance score. tDCS was not only associated with significantly better improvement in surgical performance but also did not increase error scores or rates of local discomfort compared with the sham control. However, before we could definitely apply the tDCS in surgical skill training program in the real world, future studies investigating the physiology underlying the relationship between enhancing tDCS and improvements in surgical performance should be warranted. Furthermore, future RCTs investigating different efficacy by tDCS stimulation over different cortical sites should be warranted.

## Figures and Tables

**Figure 1 brainsci-11-00707-f001:**
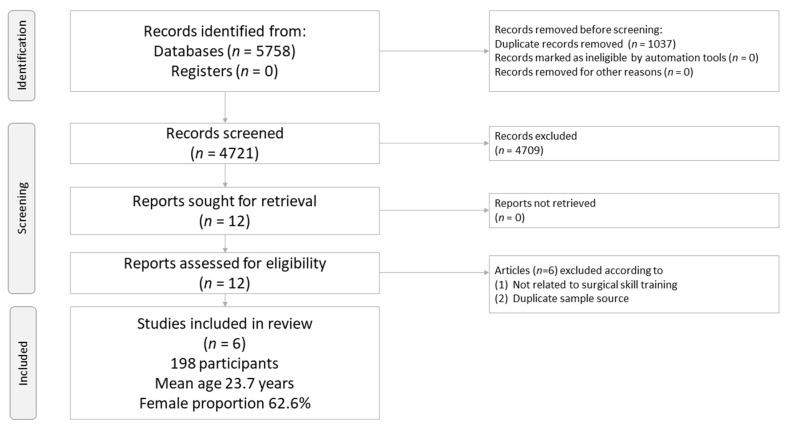
PRISMA2020 Flowchart of current network meta-analysis.

**Figure 2 brainsci-11-00707-f002:**
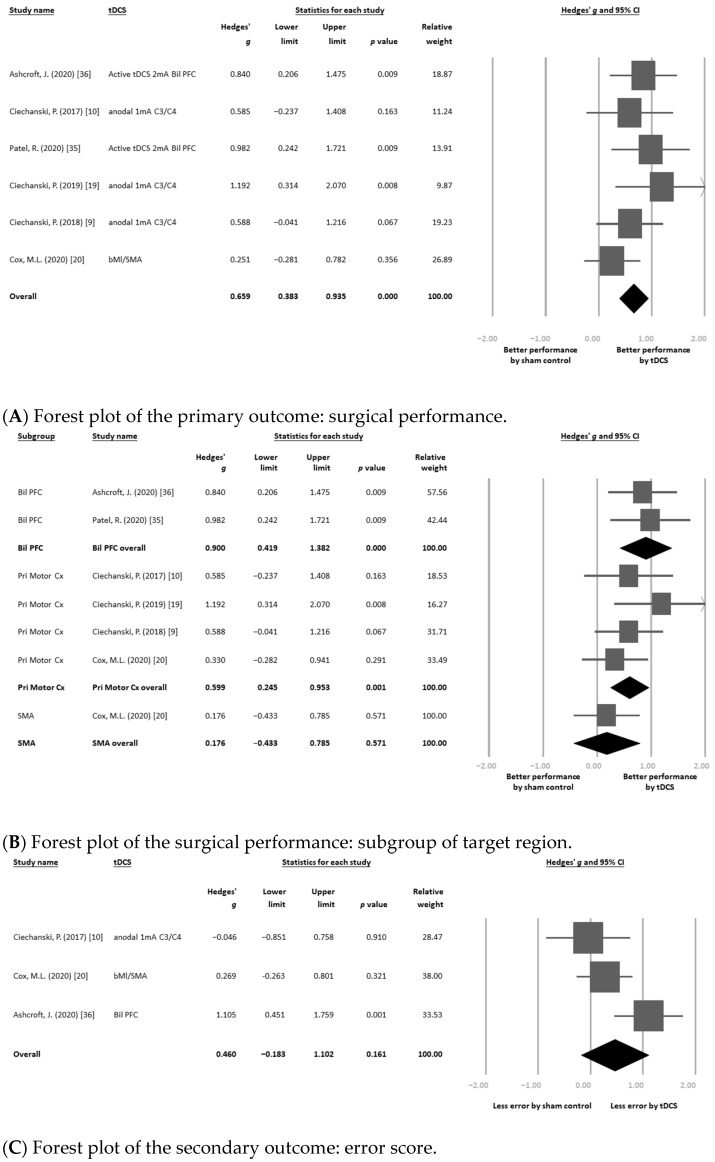
Forest plots of (**A**) the primary outcome: surgical performance; (**B**) surgical performance: subgroup of target region; (**C**) the secondary outcome: error score; (**D**) error score: subgroup of target region; and (**E**) the safety profile: rate of local discomfort. The bold rows were the overall statistical results of each subgroups.

**Table 1 brainsci-11-00707-t001:** Characteristics of the included randomized controlled trials.

Author (Year)	Task to Learn	tDCS	Comparison	Subjects	Mean Age	Female (%)	Adverse Event	Country
Ashcroft, J. (2020) [38]	knot-tying task	Anode and cathode tDCS 2 mA over F3 and F4, respectively, for 15 min	Active tDCS	20	21.3 ± 2.5	55.0	Not mentioned	Multiple countries
			Sham tDCS	20	21.9 ± 2.2	55.0		
Cox, M.L. (2020) [22]	Fundamentals of laparoscopic surgery	bM1: cathode tDCS 2 mA at right primary motor cortex + anode at left primary motor cortex over 20 min SMA: the anode over Cz and the cathode over Fpz over 20 min	Active tDCS (bM1)	20	21.9 ± 5.2	60.0	Lightheadedness, and dizziness	USA
			Active tDCS (SMA)	20	23.5 ± 5.4	80.0		
			Sham tDCS	20	22.7 ± 3.7	70.0		
Patel, R. (2020) [37]	robotic-suturing task	Active tDCS 2 mA at bilateral prefrontal cortex over 15 min	Active tDCS	15	NA	46.7	Not mentioned	Multiple countries
			Sham tDCS					
Ciechanski, P. (2019) [21] *	Fundamentals of laparoscopic surgery pattern cutting and peg transfer tasks	anode tDCS 1 mA at dominant primary motor cortex + cathode at contralateral either F3 or F4 for 20 min	Active tDCS	11	25.9 ± 3.6	72.7	Tingling, itching, and warmness	Canada
			Sham tDCS	11	25.5 ± 4.7	72.7		
Ciechanski, P. (2018) [9] *	Fundamentals of laparoscopic surgery using simulation-based task training	anode tDCS 1 mA at dominant primary motor cortex + cathode at contralateral supraorbital area over 20 min	Active tDCS	20	26.3 ± 4.1	55.0	Itching, burning, tingling, and pain	Canada
			Sham tDCS	19	24.7 ± 3.3	52.6		
Ciechanski, P. (2017) [10] *	ultrasonic aspirator to resect 3 virtual tumors embedded in healthy brain with NeuroTouch Neurosurgical simulator	anode tDCS 1 mA at dominant primary motor cortex + cathode at contralateral supraorbital area for 20 min	Active tDCS	11	25.8 ± 3.0	72.7	Itching, tingling, burning, and pain	Canada
			Sham tDCS	11	24.6 ± 2.1	72.7		

*: no duplicate sample source according to the original articles.

## Data Availability

Data of the current study was available upon reasonable request.

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
