# Peer review of "The Efficacy of Transcranial Direct Current Stimulation in Enhancing Surgical Skill Acquisition: A Preliminary Meta-Analysis of Randomized Controlled Trials"

_brainsci, 2021, doi:10.3390/brainsci11060707_

Round 1
Reviewer 1 Report
The manuscript reports a metanalysis of the effects of transcranial direct current stimulation (tDCS) on surgical skills acquisition and transfer. A very few available randomized controlled trials (RCTs) exist on the matter and were obtained from six randomized controlled trials including 198 participants. The analysis suggests an improvement in surgical performance following active compared to sham tDCS.
This work is interesting but has it several limitations some of which have been already mentioned by the author.
- The utility of the study protocol related to this metanalysis is limited by the fact that has been registered one week prior to the submission of this manuscript. In addition, when reading the protocol (/inplasy2021.4.0099), as well as the title and abstract of this paper, the reader will understand that this paper focused on tDCS. However, in the methodology of the manuscript (electronic search) and in the discussion, the authors mentioned that they also aimed to assess the effects of other neuromodulation techniques but did not to find relevant papers.
- In table 1 and throughout the manuscript, it is better to use the conventional terms that are employed to describe tDCS setups.
- For tDCS over the SMA, it is more appropriate to state “the anode over Cz and the cathode over Fpz"
- For bifrontal tDCS, it is more appropriate to state “the anode and the cathode over F3 and F4, respectively”
- when talking about the electrodes location, it is preferable to say cathode over F4 instead of cathodal over F4, and it is preferable to only use the words 'cathodal' or 'anodal' when talking about the stimulation protocol (e.g., anodal tDCS over the motor cortex).
- The statement in l. 121-127 needs to be clarified. The authors stated that the comparator is [(3) C: sham stimulation] but they continue by saying that RCTs with either sham-control or active control were included.
- Safety profile was limited to rate of local discomfort (i.e., itching, tingling pain, or erythematous). Other side-effects could be observed with tDCS (e.g., headaches, insomnia, somnolence, fatigue, phosphenes, etc.) and they merit to be addressed if they were reported in the selected studies.
- The statement in l. 162-170, needs to be clarified. For the studies [9,10,19], it is important to mention the stimulation side (anode over the predominant motor cortex and cathode over the contralateral supraorbital or prefrontal area). For study [20], it is important to explain that the cathode and the anode were placed over the right and left M1, respectively. For the study [35], the authors said that Patel and colleagues did not specify the polarity over prefrontal cortex. However, the one can refer to the article published by the same authors (Patel et al., February, 2021) who seem to put the anode and the cathode over F3/F4 respectively; this is known in tDCS literature as bifrontal tDCS. This is the same design as study [36] which was done by the same team: Patel R, Singh H, Ashcroft J, Woods AJ, Darzi A, Leff DR. Dataset of prefrontal transcranial direct-current stimulation to improve early surgical knot-tying skills. Data Brief. 2021 Feb 23;35:106905.
- The analysis and discussion do not account for the effects of several factors such as the number of tDCS sessions per RCT, the duration of sessions, the electrodes shape and size, the type of used medium (e.g., saline), the stimulation intensity, the method used to localize the cortical areas (e.g., 10-20 EEG system, Neuronavigation), the type of sham across the trials (ramping up/down or else).
- Discussion:
- SMA was targeted in only one study relative to motor and prefrontal cortices which were targeted in 2-3 studies.
- The results are limited by the fact that surgical performance varied widely among the individual RCTs.
- Statement in l. 256-266 should be paraphrased to follow the conventional way of presenting tDCS setups, e.g., anode over Cz (targeting the supplementary motor area) and cathode over Fpz.
- “without mention of the specific polarity of the tDCS” is not an issue because bifrontal tDCS (with anode over F3 and cathode over F4 is among the most used prefrontal setups in tDCS studies, especially to enhance cognitive functions or improve depression).
- Statement in l. 270-273 would benefit from paraphrasing to render it less confirmatory, in the absence of sufficient data on this matter.
- Statement in l. 359-366 merits to be clarified.
- Some editing comments: ‘mean’ female proportion (figure1 and manuscript body), motor cortex ‘stimulation’ (l. 165), electrical current (L.203, should read intensity of electrical current), bMI (table 1, should read bM1), cortical excitement (should read excitability), anodal/cathodal patches (should read electrodes), cathode at contralateral hemisphere surrounding F3/F4 (in the table, could be modified to make it clear that it is either F3 or F4). P values could be omitted from the discussion.
Author Response
Reviewer 1
The manuscript reports a metanalysis of the effects of transcranial direct current stimulation (tDCS) on surgical skills acquisition and transfer. A very few available randomized controlled trials (RCTs) exist on the matter and were obtained from six randomized controlled trials including 198 participants. The analysis suggests an improvement in surgical performance following active compared to sham tDCS.
This work is interesting but has it several limitations some of which have been already mentioned by the author.
=> Many thanks to your valuable comments. We truly appreciated your great comments and recommendation. We had rechecked the entire manuscript and made appropriate revisions on some of the text, tables, figures, and references throughout the whole manuscript according to all the reviewers’ comments. We had marked the revision with red-color in the manuscript. We believed that the revised manuscript would provide more comprehensive and scientifical information in the clinical practice.
- The utility of the study protocol related to this metanalysis is limited by the fact that has been registered one week prior to the submission of this manuscript. In addition, when reading the protocol (/inplasy2021.4.0099), as well as the title and abstract of this paper, the reader will understand that this paper focused on tDCS. However, in the methodology of the manuscript (electronic search) and in the discussion, the authors mentioned that they also aimed to assess the effects of other neuromodulation techniques but did not to find relevant papers.
=> Many thanks to your important comments. In order to provide as many information as possible, we initially intend to include all different non-invasive brain stimulation method in our literature search strategy. However, there were only randomized controlled trials of tDCS available at present time. Therefore, the MAJOR issue of the current study turned out to focus on tDCS only. Therefore, to address the changes of major focus, we had revised our statement into “…In order to provide a point of view of potentially beneficial effect by tDCS application in surgery skill acquisition, the aim of the current preliminary meta-analysis was to investigate the association between the application of tDCS and the efficacy of surgical performance during surgical skill training. Furthermore, in addition to the overall result of tDCS application, we made subgroup analysis based on the different stimulation cortical sites in order to determine the potentially different effects by the different stimulation sites. Finally, in order to provide as many information as possible, we also intended to include all different non-invasive brain stimulation method in our literature search strategy…” in the section of introduction (page 3 line 103-110) and added “…There were no any RCTs investigating the efficacy of other non-invasive brain stimulation on surgical skill training program…” in the section of result (page 4 line 185-186) accordingly.
- In table 1 and throughout the manuscript, it is better to use the conventional terms that are employed to describe tDCS setups.
- For tDCS over the SMA, it is more appropriate to state “the anode over Cz and the cathode over Fpz"
=> Many thanks to your excellent comments. We had revised our statement of “tDCS over SMA” into “the anode over Cz and the cathode over Fpz” throughout the manuscript accordingly.
- For bifrontal tDCS, it is more appropriate to state “the anode and the cathode over F3 and F4, respectively”
=> Many thanks to your great comments. We had revised our statement of “bifrontal tDCS” into “Anode and cathode tDCS 2mA over F3 and F4, respectively” for the statement of Ashcroft, J. (2020)[36] in table 1 accordingly (page 5 Table 1). However, because there was no detailed information about the specific direction of current in Patel, R. (2020)[35] and Ciechanski, P. (2019)[19], we could not change the statement of tDCS into this specific statement.
- when talking about the electrodes location, it is preferable to say cathode over F4 instead of cathodal over F4, and it is preferable to only use the words 'cathodal' or 'anodal' when talking about the stimulation protocol (e.g., anodal tDCS over the motor cortex).
=> Many thanks to your valuable suggestion. We had changed these spelling from “anodal / cathodal” into “anode / cathode” throughout the manuscript accordingly.
- The statement in l. 121-127 needs to be clarified. The authors stated that the comparator is [(3) C: sham stimulation] but they continue by saying that RCTs with either sham-control or active control were included.
=> Many thanks to your important suggestion. We had revised our PICO setting into “…(1) P: participants receiving surgical skill training; (2) I: transcranial direct-current stimulation (tDCS); (3) C: sham stimulation or active control; and (4) O: the change in surgical performance…”accordingly (page 3 line 130-132).
- Safety profile was limited to rate of local discomfort (i.e., itching, tingling pain, or erythematous). Other side-effects could be observed with tDCS (e.g., headaches, insomnia, somnolence, fatigue, phosphenes, etc.) and they merit to be addressed if they were reported in the selected studies.
=> Many thanks to your excellent suggestion. We had added detailed information about adverse events reported in each study accordingly. In addition, we had added one statement “…To be specific, the most frequently reported adverse event included tingling, burning sensation, itching, and pain (table 1)…” in the section of result accordingly (page 7 line 256-258).
- The statement in l. 162-170, needs to be clarified. For the studies [9,10,19], it is important to mention the stimulation side (anode over the predominant motor cortex and cathode over the contralateral supraorbital or prefrontal area). For study [20], it is important to explain that the cathode and the anode were placed over the right and left M1, respectively. For the study [35], the authors said that Patel and colleagues did not specify the polarity over prefrontal cortex. However, the one can refer to the article published by the same authors (Patel et al., February, 2021) who seem to put the anode and the cathode over F3/F4 respectively; this is known in tDCS literature as bifrontal tDCS. This is the same design as study [36] which was done by the same team: Patel R, Singh H, Ashcroft J, Woods AJ, Darzi A, Leff DR. Dataset of prefrontal transcranial direct-current stimulation to improve early surgical knot-tying skills. Data Brief. 2021 Feb 23;35:106905.
=> Many thanks to your great suggestion. We had added those detailed information as “…Among these six eligible RCTs, three investigated the efficacy of anodal tDCS over dominant primary motor cortex/cathodal tDCS over contralateral hemisphere surrounding F3/F4 for 20 minutes [9,10,19], one investigated the efficacy of cathodal tDCS over right primary motor cortex (M1) stimulation and anode over left primary motor cortex (M1) [20]…”according to your great suggestion (page 4 line 174-178). However, about the new reference (Patel et al., February, 2021), this new article was the detailed data of Ashcroft, J. (2020) [36]. In the original statement of Patel et al., February, 2021, the authors said “The dataset reported here was collected from a double-blind randomized sham-controlled trial investigating the performance enhancing effects of prefrontal transcranial direct-current stimulation (tDCS) on surgical knot-tying performance [7] (Ref 7: J. Ashcroft , R. Patel , A.J. Woods , A. Darzi , H. Singh , D.R. Leff, Prefrontal transcranial direct-current stimulation im- proves early technical skills in surgery, Brain Stimul. (2020) In Review)”. Therefore, the detailed information of Patel, R. (2020) [35] was still unspecified. To address this, we revised our statement as “…one investigated the efficacy of tDCS without detailed information about the specific polarity over prefrontal cortex stimulation [35]…” (page 4 line 178-179).
- The analysis and discussion do not account for the effects of several factors such as the number of tDCS sessions per RCT, the duration of sessions, the electrodes shape and size, the type of used medium (e.g., saline), the stimulation intensity, the method used to localize the cortical areas (e.g., 10-20 EEG system, Neuronavigation), the type of sham across the trials (ramping up/down or else).
=> Many thanks to your valuable comment. The heterogeneity of the study design (and protocol application) among the included studies were one of the potential bias in the current study. However, the relatively small numbers of the included RCTs limited our space to make further subgroup analysis and further discussion. Therefore, we had added these potential source of heterogeneity in the section of limitation as “…Finally, the study design and tDCS protocols applied in the included RCTs were widely heterogeneous (i.e. number of tDCS sessions, duration of each sessions, electrodes shape/size, type of medium used (for example, normal saline), stimulation intensity, method used to localize the cortical areas (for example 10-20 EEG system or Neuronavigation), and type of sham)…” (page 11 line 426-431).
- Discussion:
- SMA was targeted in only one study relative to motor and prefrontal cortices which were targeted in 2-3 studies.
=> Many thanks to your important comment. We had added the statement of “…The number of the RCT investigating the efficacy of tDCS over the bilateral motor cortex or supplementary motor area on the unimanual/bimanual function was relatively smaller than the other subgroups. Therefore, clinicians should pay special attention when applied our result in their surgical training program…” in the section of discussion accordingly (page 11 line 382-386).
- The results are limited by the fact that surgical performance varied widely among the individual RCTs.
=> Many thanks to your excellent comment. The heterogeneity of the simulated surgical performance was one of the limitation of the current meta-analysis. Therefore, we had addressed this limitation as “…Second, a wide variety of surgical simulation and variety of evaluation scales for surgical performance measurement, such as pattern-cutting scores, percent of changes in the amount of tumor resected, knot tensile strength, and overall performance score, was applied among the included RCTs. The choice of the surgical performance measurement in each RCT varied along with the surgery type (i.e., neurosurgery or laparoscopic surgery) and the method of surgery simulation. Although we could not perform specific subgroup analyses based on specific surgical performance measurement, there had been significantly improvement in pattern-cutting scores, percent of changes in the amount of tumor resected, knot tensile strength, and overall performance score according to the result of their original RCTs. Although there was no statistically significant heterogeneity noted in the current study, the clinicians should pay special attention when applying the results of the current study in their surgical training program…” in the section of limitation accordingly (page 11 line 395-406).
- Statement in l. 256-266 should be paraphrased to follow the conventional way of presenting tDCS setups, e.g., anode over Cz (targeting the supplementary motor area) and cathode over Fpz.
=> Many thanks to your great suggestion. We had changed these spelling from “anodal / cathodal” into “anode / cathode” throughout the manuscript accordingly.
- “without mention of the specific polarity of the tDCS” is not an issue because bifrontal tDCS (with anode over F3 and cathode over F4 is among the most used prefrontal setups in tDCS studies, especially to enhance cognitive functions or improve depression).
=> Many thanks to your valuable comment. In the most tDCS studies, the anode over F3 and cathode over F4 were the most used prefrontal setups in tDCS studies in neuropsychiatric study to enhance the cognitive function. Among the most included RCTs in the current meta-analysis, they followed this rationale, too. However, as reply in the previous comment, the new reference (Patel et al., February, 2021) was the detailed data of Ashcroft, J. (2020) [36]. In the original statement of Patel et al., February, 2021, the authors said “The dataset reported here was collected from a double-blind randomized sham-controlled trial investigating the performance enhancing effects of prefrontal transcranial direct-current stimulation (tDCS) on surgical knot-tying performance [7] (Ref 7: J. Ashcroft , R. Patel , A.J. Woods , A. Darzi , H. Singh , D.R. Leff, Prefrontal transcranial direct-current stimulation im- proves early technical skills in surgery, Brain Stimul. (2020) In Review)”. Therefore, the detailed information of Patel, R. (2020) [35] was still unspecified. To address this, we had revised our statement as “…tDCS 2 mA targeted at the bilateral prefrontal cortex without detailed information about the specific polarity of the tDCS [35], and anodal tDCS over left dorsolateral prefrontal cortex (DLPFC) (F3) plus cathodal tDCS over right DLPFC (F4) [36]…” (page 9 line 291-294).
- Statement in l. 270-273 would benefit from paraphrasing to render it less confirmatory, in the absence of sufficient data on this matter.
=> Many thanks to your important comment. We had revised our statement into a less confirmatory one as “…To be specific, although there was inconclusive evidence the low-current (i.e. 1 mA) cathodal tDCS stimulation might contribute to the suppressing effect [37], the high-current (i.e. 2 mA) cathodal tDCS stimulation might be associated with excitatory changes [38]. Therefore, all the investigated tDCS stimulations included in the current study might all have enhancing effects on the targeted cortex (i.e., anodal tDCS 1 mA or cathodal tDCS 2 mA)…” accordingly (page 9 line 296-302).
- Statement in l. 359-366 merits to be clarified.
=> Many thanks to your great comment. We had extended the discussion about neural noise as “…In addition, the different targeted cortices would yield different results (i.e., targeted to the primary motor cortex vs to the supplementary motor area). There had been argument about the different cognition effect of tDCS on differently targeted cortex [42]. However, in the previous review of computational neuroscience modeling studies on stochastic resonance, the “neural noise” produced by tDCS, either in forms of depolarization or hyperpolarization in different cortex, can indeed improve cognitive performance [42,43]. To be specific, according to the neural noise hypothesis, the after-effect of tDCS might depend on the overall glutamatergic, GABAergic, dopaminergic, and serotoninergic synaptic activity. Therefore, the improvement of cognition should not be simply attributed to “better activity, better function” [42]. Therefore, the potentially beneficial effect of tDCS on cognitive performance might not be simply explained by the polarity or targeted cortex of tDCS. Future RCTs investigating different efficacy on surgical skill acquisition by tDCS over different targeted cortex should be warranted…” accordingly (page 10 line 358-371).
- Some editing comments: ‘mean’ female proportion (figure1 and manuscript body), motor cortex ‘stimulation’ (l. 165), electrical current (L.203, should read intensity of electrical current), bMI (table 1, should read bM1), cortical excitement (should read excitability), anodal/cathodal patches (should read electrodes), cathode at contralateral hemisphere surrounding F3/F4 (in the table, could be modified to make it clear that it is either F3 or F4). P values could be omitted from the discussion.
=> Many thanks to your excellent comment. We had revised all the typos accordingly and deleted the ‘mean’ female proportion and motor cortex ‘stimulation’ accordingly.
Reviewer 2 Report
This meta-analysis covers a very interesting topic, i.e. the possibility of enhancing the surgical training by means of tDCS. My major concern is that, in my opinion, the field is very young and the effort of performing a meta-analysis and drawing any conclusion, even tentative, may result over-ambitious. In fact, the authors retrieved only six eligible papers, three of which are from the same research group and one of the remaining three is a poster from a meeting. Moreover, these few studies are very different from each other for many important features: they use five different tDCS montages and they target at least four different surgical skills, which in turn involve as many different cognitive, motor and visuo-spatial functions. Actually, the authors acknowledge these limitations at the end of the discussion but, in my opinion, the should go more in this aspect in order to resize the importance of the variability among studies and make more credible the idea that tDCS is generically efficacious in improving the surgical skills. One good start point for this additional discussion might be the concept of “neural noise”, that they only mention, which could explain a broad and non-specific beneficial effect.
Minor points are:
- In materials and methods the electronic searches with the keywords DBS, TMS, TBS and VNS come unexpected because neither in the title nor in the abstract and in the introduction were mentioned stimulation techniques other than tDCS. Therefore, the authors should either remove this part from the methods section, or mention this aim of their study also in the previous sections.
- In the discussion, the sentence at the lines 332-334 “However, the small numbers of included RCTs in the current meta-analysis were contributed to the difficulty to recruit large scale surgeons to join in such project” is not clear to me. Maybe it should be rephrased.
- At line 376 should be “investigating” and not “investigated”
- In the final discussion (or conclusion) the authors could also mention that the possible improvement of surgical performance by means of tDCS, if definitively proved, could impact surgical practice even after the training completion, and the tDCS could become a regular work instrument for surgeons.
Author Response
Reviewer 2
This meta-analysis covers a very interesting topic, i.e. the possibility of enhancing the surgical training by means of tDCS. My major concern is that, in my opinion, the field is very young and the effort of performing a meta-analysis and drawing any conclusion, even tentative, may result over-ambitious. In fact, the authors retrieved only six eligible papers, three of which are from the same research group and one of the remaining three is a poster from a meeting. Moreover, these few studies are very different from each other for many important features: they use five different tDCS montages and they target at least four different surgical skills, which in turn involve as many different cognitive, motor and visuo-spatial functions. Actually, the authors acknowledge these limitations at the end of the discussion but, in my opinion, the should go more in this aspect in order to resize the importance of the variability among studies and make more credible the idea that tDCS is generically efficacious in improving the surgical skills.
=> Many thanks to your valuable comments. We truly appreciated your great comments and recommendation. We had rechecked the entire manuscript and made appropriate revisions on some of the text, tables, figures, and references throughout the whole manuscript according to all the reviewers’ comments. We had marked the revision with red-color in the manuscript. The current study aimed to provide a tentative view of new method to enhance the surgical skill training program. As addressed in the discussion and limitation, the heterogeneity of the study design (and protocol application) among the included studies were one of the potential bias in the current study. However, the relatively small numbers of the included RCTs limited our space to make further subgroup analysis and further discussion. Therefore, to address this, we had added these potential source of heterogeneity as “…Finally, the study design and tDCS protocols applied in the included RCTs were widely heterogeneous (i.e. number of tDCS sessions, duration of each sessions, electrodes shape/size, type of medium used (for example, normal saline), stimulation intensity, method used to localize the cortical areas (for example 10-20 EEG system or Neuronavigation), and type of sham)…”in the section of limitation (page 11 line 426-431) and the statement of “…The number of the RCT investigating the efficacy of tDCS over the bilateral motor cortex or supplementary motor area on the unimanual/bimanual function was relatively smaller than the other subgroups. Therefore, clinicians should pay special attention when applied our result in their surgical training program…” in the section of discussion accordingly (page 11 line 382-386). In addition, to provide more detailed information about the merits and demerits of the tDCS, we had added detailed information about adverse events reported in each study accordingly. In addition, we had added one statement “…To be specific, the most frequently reported adverse event included tingling, burning sensation, itching, and pain (table 1)…” in the section of result accordingly (page 7 line 256-258). We believed that the revised manuscript would provide more balanced, comprehensive, and scientifical information in the clinical practice.
One good start point for this additional discussion might be the concept of “neural noise”, that they only mention, which could explain a broad and non-specific beneficial effect.
=> Many thanks to your great comment. We had extended the discussion about neural noise as “…In addition, the different targeted cortices would yield different results (i.e., targeted to the primary motor cortex vs to the supplementary motor area). There had been argument about the different cognition effect of tDCS on differently targeted cortex [42]. However, in the previous review of computational neuroscience modeling studies on stochastic resonance, the “neural noise” produced by tDCS, either in forms of depolarization or hyperpolarization in different cortex, can indeed improve cognitive performance [42,43]. To be specific, according to the neural noise hypothesis, the after-effect of tDCS might depend on the overall glutamatergic, GABAergic, dopaminergic, and serotoninergic synaptic activity. Therefore, the improvement of cognition should not be simply attributed to “better activity, better function” [42]. Therefore, the potentially beneficial effect of tDCS on cognitive performance might not be simply explained by the polarity or targeted cortex of tDCS. Future RCTs investigating different efficacy on surgical skill acquisition by tDCS over different targeted cortex should be warranted…” accordingly (page 10 line 358-371).
Minor points are:
- In materials and methods the electronic searches with the keywords DBS, TMS, TBS and VNS come unexpected because neither in the title nor in the abstract and in the introduction were mentioned stimulation techniques other than tDCS. Therefore, the authors should either remove this part from the methods section, or mention this aim of their study also in the previous sections.
=> Many thanks to your important comments. In order to provide as many information as possible, we initially intend to include all different non-invasive brain stimulation method in our literature search strategy. However, there were only randomized controlled trials of tDCS available at present time. Therefore, the MAJOR issue of the current study turned out to focus on tDCS only. Therefore, to address the changes of major focus, we had revised our statement into “…In order to provide a point of view of potentially beneficial effect by tDCS application in surgery skill acquisition, the aim of the current preliminary meta-analysis was to investigate the association between the application of tDCS and the efficacy of surgical performance during surgical skill training. Furthermore, in addition to the overall result of tDCS application, we made subgroup analysis based on the different stimulation cortical sites in order to determine the potentially different effects by the different stimulation sites. Finally, in order to provide as many information as possible, we also intended to include all different non-invasive brain stimulation method in our literature search strategy…” in the section of introduction (page 3 line 103-110) and added “…There were no any RCTs investigating the efficacy of other non-invasive brain stimulation on surgical skill training program…” in the section of result (page 4 line 185-186) accordingly.
- In the discussion, the sentence at the lines 332-334 “However, the small numbers of included RCTs in the current meta-analysis were contributed to the difficulty to recruit large scale surgeons to join in such project” is not clear to me. Maybe it should be rephrased.
=> Many thanks to your excellent comments. We had revised this statement into “…However, the difficulty to recruit large numbers of surgeons to join in such RCT would limit the overall numbers of such kinds of RCTs. Therefore, it would be difficult to recruit large numbers of RCTs in the meta-analysis investigating interventions to enhance surgical skill training program…” accordingly (page 11 line 390-395).
- At line 376 should be “investigating” and not “investigated”
=> Many thanks to your valuable suggestion. We had revised this typo into “…Also, future RCTs investigating different efficacy by tDCS stimulation over different cortical sites should be warranted…” accordingly (page 12 line 441-443).
- In the final discussion (or conclusion) the authors could also mention that the possible improvement of surgical performance by means of tDCS, if definitively proved, could impact surgical practice even after the training completion, and the tDCS could become a regular work instrument for surgeons.
=> Many thanks to your great suggestion. The tDCS had been approved to improve cognition in many different neuropsychiatric researches. However, because the current meta-analysis aimed to provide a tentative direction of new surgical skill training program, we should not make a conclusion in this stage. Rather, we would like to recommend future large-scale RCTs to support our result. Therefore, we had revised our statement into “…However, before we could definitely apply the tDCS in surgical skill training program in the real world, future studies investigating the physiology underlying the relationship between enhancing tDCS and improvements in surgical performance should be warranted. Also, future RCTs investigating different efficacy by tDCS stimulation over different cortical sites should be warranted…” in the section of conclusion accordingly (page 12 line 438-443).
Reviewer 3 Report
The manuscripts presents a meta-analysis of an interesting application of tDCS: the improvement of surgical skill learning. Clearly this has practical implications, but also there are potentially implications for understanding how tDCS enhances a complex motor skill in a healthy adult population.
Some improvements should be made to the background literature:
1) The literature most clearly relevant to tDCS as applied to surgical skill learning is when tDCS is applied to improve motor learning of manual tasks. However this literature is not adequately referenced, and instead the authors pick a reference that looks at balance control (ref 18, L 88 and L 281). Far more relevant is the extensive work done looking at alterations in manual skill as a result of tDCS. For instance changes on the Jebsen-Taylor Hand Function Test, alterations in finger sequencing, serial reaction time tasks, maximal pinch force etc. Indeed an inclusion of this literature could have led to a more interesting manuscript as the authors could make suggestions as to which of these experimental paradigms/clinical tests more closely aligned to the surgical skill training. This could then be followed up with a discussion of potential differential impact of tDCS on motor skills. Examples of relevant background literature could include from Reis et al , Stagg et al, Karok et al and many others.
2) The authors allude to the reported non-linearity of tDCS effects both in the introduction and discussion. This is an important observation in the literature but there is still uncertainty in this area. A relevant reference here would be Esmaeilpour et al (2018).
3) The materials and method section mention that the authors were also seeking to include other NIBS methods but were unable to find suitable RCTs in the literature (L105-L115). This is fine to exclude, but these NIBS all have very different physiological mechanisms and so if they are to be mentioned then this should be covered in the introduction to the manuscript. As it is their inclusion is quite unexpected after the introduction that is totally focused on tDCS.
4) tDCS is not just thought to influence underlying cortex but influence network activity thus leading to potentially differential impacts on different tasks etc. This should be included and expanded on in the discussion rather than a vague mention of stochastic influences. (L360)
Author Response
Reviewer 3
The manuscripts presents a meta-analysis of an interesting application of tDCS: the improvement of surgical skill learning. Clearly this has practical implications, but also there are potentially implications for understanding how tDCS enhances a complex motor skill in a healthy adult population.
=> Many thanks to your valuable comments. We truly appreciated your great comments and recommendation. We had rechecked the entire manuscript and made appropriate revisions on some of the text, tables, figures, and references throughout the whole manuscript according to all the reviewers’ comments. We had marked the revision with red-color in the manuscript. We believed that the revised manuscript would provide more comprehensive and scientifical information in the clinical practice.
Some improvements should be made to the background literature:
1) The literature most clearly relevant to tDCS as applied to surgical skill learning is when tDCS is applied to improve motor learning of manual tasks. However this literature is not adequately referenced, and instead the authors pick a reference that looks at balance control (ref 18, L 88 and L 281). Far more relevant is the extensive work done looking at alterations in manual skill as a result of tDCS. For instance changes on the Jebsen-Taylor Hand Function Test, alterations in finger sequencing, serial reaction time tasks, maximal pinch force etc. Indeed an inclusion of this literature could have led to a more interesting manuscript as the authors could make suggestions as to which of these experimental paradigms/clinical tests more closely aligned to the surgical skill training. This could then be followed up with a discussion of potential differential impact of tDCS on motor skills. Examples of relevant background literature could include from Reis et al , Stagg et al, Karok et al and many others.
=> Many thanks to your excellent comments. In order to improve the adequacy of reference, we had changed the Ref 18. Into the review article by Reis (Reis, J.; Fritsch, B. Modulation of motor performance and motor learning by transcranial direct current stimulation. Curr Opin Neurol 2011, 24, 590-596, doi:10.1097/WCO.0b013e32834c3db0) accordingly. In addition, we had added Ref 44. And Ref 45. (Ref 44: Stagg, C.J.; Jayaram, G.; Pastor, D.; Kincses, Z.T.; Matthews, P.M.; Johansen-Berg, H. Polarity and timing-dependent effects of transcranial direct current stimulation in explicit motor learning. Neuropsychologia 2011, 49, 800-804, doi:10.1016/j.neuropsychologia.2011.02.009; Ref 45: Karok, S.; Fletcher, D.; Witney, A.G. Task-specificity of unilateral anodal and dual-M1 tDCS effects on motor learning. Neuropsychologia 2017, 94, 84-95, doi:10.1016/j.neuropsychologia.2016.12.002). Furthermore, to enhance the detailed information in the introduction, we had revised our statement into “…For example, the previous articles had summarized the evidence of tDCS efficacy in motor skill training (either in fine movement or reaction time) in stroke patients and healthy subjects [18,44]. To be specific, Karok and the colleagues had demonstrated different electrode montage-dependent improvement in three experimental tasks (i.e. Purdue Pegboard test, visuomotor grip force tracking task, and visuomotor wrist rotation speed control task) by tDCS application in healthy young adults [45]…” (page 2 line 85-93) and revised our discussion as “…Similarly, the application of tDCS enhancement over participants’ primary motor cortex was found to be associated with significant improvements in balance control in stroke patients and healthy subjects [18,21]. In the study by Karok, the tDCS stimulation over M1, either right M1 or bilateral M1, significantly improved three fine movement-related task, including Purdue Pegboard test, visuomotor grip force tracking task, and visuomotor wrist rotation speed control task, compared to sham control [45]. All these key components (i.e. executive function, motor-learning function, fine movement, and reaction time) were the major part of fine movement skill acquisition; therefore, the improvement of these key components would support the rationale of the application of enhancing tDCS to improve motor skill acquisition (i.e., surgical skill acquisition) [10,39]…” (page 9 line 306-317).
2) The authors allude to the reported non-linearity of tDCS effects both in the introduction and discussion. This is an important observation in the literature but there is still uncertainty in this area. A relevant reference here would be Esmaeilpour et al (2018).
=> Many thanks to your great comments. In order to make the discussion about the non-linearity of tDCS effects more clearly, we had added the Ref 46 (Esmaeilpour, Z.; Marangolo, P.; Hampstead, B.M.; Bestmann, S.; Galletta, E.; Knotkova, H.; Bikson, M. Incomplete evidence that increasing current intensity of tDCS boosts outcomes. Brain stimulation 2018, 11, 310-321, doi:10.1016/j.brs.2017.12.002) and added further discussion as “…Another issue regarding the efficacy of tDCS application in skill training program was the “actual effect” of the applied tDCS on brain excitability. For example, the applied current intensity of tDCS would be one major argument. To be specific, the argument of the enhancing/suppressing effect to the target cortex by different current intensity (i.e. low-current versus high-current) had become one inconclusive issue [37,38]. Further, in the previous review article by Esmaeilpour and the colleague, the authors analyzed the does-response association of tDCS intensity and found a complex state-dependent non-monotonic dose response according to human neurophysiology studies [46]. This finding would further complicate the studies of tDCS in human brain function…” accordingly (page 10 line 350-358).
3) The materials and method section mention that the authors were also seeking to include other NIBS methods but were unable to find suitable RCTs in the literature (L105-L115). This is fine to exclude, but these NIBS all have very different physiological mechanisms and so if they are to be mentioned then this should be covered in the introduction to the manuscript. As it is their inclusion is quite unexpected after the introduction that is totally focused on tDCS.
=> Many thanks to your important comments. In order to provide as many information as possible, we initially intend to include all different non-invasive brain stimulation method in our literature search strategy. However, there were only randomized controlled trials of tDCS available at present time. Therefore, the MAJOR issue of the current study turned out to focus on tDCS only. Therefore, to address the changes of major focus, we had revised our statement into “…In order to provide a point of view of potentially beneficial effect by tDCS application in surgery skill acquisition, the aim of the current preliminary meta-analysis was to investigate the association between the application of tDCS and the efficacy of surgical performance during surgical skill training. Furthermore, in addition to the overall result of tDCS application, we made subgroup analysis based on the different stimulation cortical sites in order to determine the potentially different effects by the different stimulation sites. Finally, in order to provide as many information as possible, we also intended to include all different non-invasive brain stimulation method in our literature search strategy…” in the section of introduction (page 3 line 103-110) and added “…There were no any RCTs investigating the efficacy of other non-invasive brain stimulation on surgical skill training program…” in the section of result (page 4 line 185-186) accordingly.
4) tDCS is not just thought to influence underlying cortex but influence network activity thus leading to potentially differential impacts on different tasks etc. This should be included and expanded on in the discussion rather than a vague mention of stochastic influences. (L360)
=> Many thanks to your valuable suggestions. In order to expand our discussion, we had expanded this discussion as “…In addition, the different targeted cortices would yield different results (i.e., targeted to the primary motor cortex vs to the supplementary motor area). There had been argument about the different cognition effect of tDCS on differently targeted cortex [42]. However, in the previous review of computational neuroscience modeling studies on stochastic resonance, the “neural noise” produced by tDCS, either in forms of depolarization or hyperpolarization in different cortex, can indeed improve cognitive performance [42,43]. To be specific, according to the neural noise hypothesis, the after-effect of tDCS might depend on the overall glutamatergic, GABAergic, dopaminergic, and serotoninergic synaptic activity. Therefore, the improvement of cognition should not be simply attributed to “better activity, better function” [42]. Therefore, the potentially beneficial effect of tDCS on cognitive performance might not be simply explained by the polarity or targeted cortex of tDCS. Future RCTs investigating different efficacy on surgical skill acquisition by tDCS over different targeted cortex should be warranted…” (page 10 line 358-371) accordingly.
Round 2
Reviewer 1 Report
There are no further comments.
Reviewer 2 Report
The authors have thoroughly addressed all the points raised in my previous review. Therefore, in my opinion, the paper is now suitable for publication